# Revisit Weakly-Supervised Audio-Visual Video Parsing from the Language Perspective

**Yingying Fan, Yu Wu, Bo Du, Yutian Lin**[*]
School of Computer Science, Hubei Luojia Laboratory, Wuhan University
`{fanyingying_cs, wuyucs, dubo, yutian.lin}@whu.edu.cn`

## Abstract

We focus on the weakly-supervised audio-visual video parsing task (AVVP), which aims to identify and locate all the events in audio and visual modalities. Previous works only concentrate on video-level overall label denoising across modalities, but overlook the segment-level label noise, where adjacent video segments (*i.e.*, 1-second video clips) may contain different events. However, recognizing events in the segment is challenging because its label could be any combination of events that occur in the video. To address this issue, we consider tackling AVVP from the language perspective, since language could freely describe how various events appear in each segment beyond fixed labels. Specifically, we design language prompts to describe all cases of event appearance for each video. Then, the similarity between language prompts and segments is calculated, where the event of the most similar prompt is regarded as the segment-level label. In addition, to deal with the mislabeled segments, we propose to perform dynamic re-weighting on the unreliable segments to adjust their labels. Experiments show that our simple yet effective approach outperforms state-of-the-art methods by a large margin. Code and data are available at https://github.com/fyyCS/LSLD.

## 1  Introduction

Classic audio-visual tasks such as audio-visual event localization and audio-visual representation learning have been well explored in previous works [1, 2, 3] and have achieved promising performance. However, most existing research studies on these tasks assume that audio and visual are correlated and temporally aligned, which is not always the case. For example, a person can hear a dog barking but not see it in the video. To this end, we focus on the more general audio-visual video parsing (AVVP) task, which aims to recognize all events in each modality and localize their temporal boundaries.

Since the full annotation process is time-consuming and expensive, Tian *et al.* [4] address AVVP in a weakly-supervised manner, which only requires video-level labels, without specifying the modality and temporal boundaries. They develop a hybrid attention network to leverage unimodal and cross-modal information and propose a simple Multimodal Multiple Instance Learning (MMIL) framework to aggregate segment-level predictions into video-level ones. However, weakly-supervised labels cannot determine which modality the event comes from. Wu *et al.* [5] then introduce a video-level label refinement method and adopt contrastive learning for audio and visual temporal alignment. Furthermore, Chen *et al.* [6] propose another label denoising method based on alleviating modality-specific noise through cross-modal loss patterns for both modalities.

However, previous works [5, 6] only considered the noisy label at the overall video instance level, while ignoring the **segment-level label noise**. Segment-level label noise indicates the fact that *not every segment of a video contains all the events of the video*. For example, a person may be *speeching*

---

[*]Corresponding author

in the video for a few seconds, and then he disappears in the following seconds. Therefore, it is inappropriate to consider *speech* as a supervised signal for all segments, where *speech* can be regarded as a noisy label for some segments of this video. It could harm the network in event classification on segments when training with such segment-level noisy labels.

The challenge of learning reliable segment-level labels comes from that the label of each segment is not predefined, but could be any combination of events that occur in the video, *e.g.*, a segment may contain both speech and music events or none of them. To address this flexible label assignment challenge, we propose to leverage a third modality, the language, which enables better scene understanding beyond the labels. Language can describe what is happening in the scene, and who is involved, and identify segments not covered by the existing labels. Therefore, we propose to address AVVP from the language perspective, where natural language descriptions serve as a bridge connecting video and flexible weak labels.

In this paper, we propose a **Language guided Segment-level Label Denoising (LSLD)** strategy. We leverage the generalization capability of language to construct prompts that indicate all cases of event appearance for a segment. To connect vision and language, we introduce a vision-language pre-trained model, CLIP [7], into our work. Specifically, CLIP is used to extract the visual features for all segments, and the similarity between the segment and all prompts is calculated for label denoising. In addition, to further mitigate the effect of noisy segment labels, we perform a dynamic re-weighting strategy on the segments that may be mislabeled. Specifically, we calculate the similarity between the language description of each event and all the segments across the dataset. We do this for both the segments with or without the event label (*i.e.*, potentially positive and negative clips) separately to compare their similarity distribution. If a segment's similarity falls within the overlap of the positive and negative groups, we consider it an unreliable segment and adjust its label dynamically according to its similarity to the event.

For the audio track, we introduce a large-scale audio-language pre-trained model LAION-CLAP [8] to denoise the audio label. The main contributions of our work are summarized as follows:

- We introduce the language modality to the AVVP task and construct language prompts that indicate all cases of event appearance for a segment, where the prompt with the highest similarity is regarded as the segment label to avoid segment-level label noise.

- We propose a simple yet effective Language-guided Segment-level Label Denoising (LSLD) mechanism based on CLAP and CLIP to align the modality of audio/visual and language for segment-level label denoising. In addition, we mine the visual segments with unreliable labels and learn from them with dynamic weight.

- Experiments show our method achieves comparable performance with the state-of-the-art methods. Especially, we improve visual metric significantly from 66.4% to 71.3% on the segment level over the state-of-the-art.

## 2   Related work

**Audio-Visual Learning.** Audio-visual learning includes various interesting tasks such as audio-visual representation learning [9, 10, 11, 12, 13, 14, 3, 15], audio-visual event localization [1, 16, 17, 18, 19, 20, 21], audio-visual sound separation [22, 23, 24, 25, 26, 27], audio-visual sound source localization [28, 29, 30, 31], and audio-visual action recognition [32, 33, 34, 35]. Most previous work on these tasks assumes that audio and visual are temporally aligned and always correlated. However, in many videos, audio and visual do not share the same information at a certain time, for instance, something that can be seen is not audible and the sound source is not visible. Therefore, we focus on the audio-visual video parsing task and aim to separately identify the events that happen in each modality with only weakly-supervised video-level labels for training.

**Audio-Visual Video Parsing.** Audio-visual video parsing (AVVP) task [4] intends to recognize events in each modality and localize their temporal boundaries. Since labeling each modality and segment of a video is time-consuming and labor-intensive, recent work has been conducted in a weakly-supervised manner. Tian *et al.* [4] utilize a hybrid attention network and multiple instance learning mechanism to aggregate segment features into video-level ones. Wu *et al.* [5] adopt the exchange of unrelated video and audio tracks to denoise weakly-supervised labels at the video level. More recently, to explicitly group semantic-aware multi-modal contexts, Mo *et al.* [36] propose

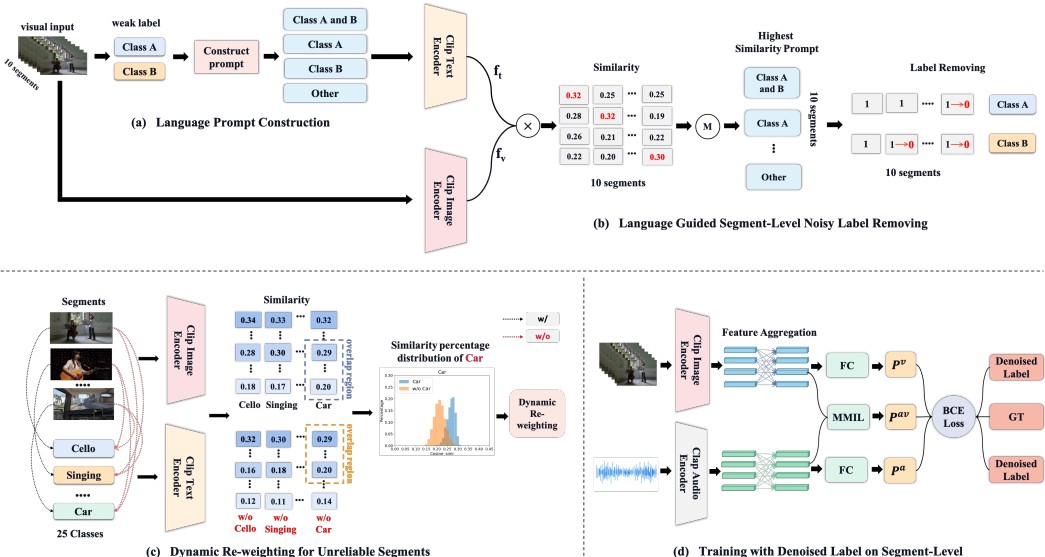

Figure 1: Algorithm Overview. (a) **Language prompts construction** provides prompts for a video to indicate all cases of event appearance based on the weak label of the video. (b) **Language guided segment-Level noisy-label removing** calculates the similarity between prompts and segments, where the prompt with the highest similarity is taken as the segment label. (c) **Dynamic re-weighting for unreliable segments** gathers the segments with and without the event using the denoised labels and calculates their similarity to the event name respectively, then re-weight the segments' label on the overlap region according to their similarity. (d) **Training with Denoised Label on Segment-Level**. Finally, with denoised segment-level labels, we train the network following [4]. CLIP and CLAP are frozen during the process. In the figure, "$\rightarrow 0$" denotes label removal, $\otimes$ is the cross product, and Ⓜ denotes the maximum function.

a Multi-modal Grouping Network to learn more discriminative audio and visual representations. Chen *et al.* [6] propose a dynamic training mechanism, which lowers the modality-specific noise at the video level. Different from the above approaches, we propose to investigate segment-level label noise. Specifically, we introduce the language modality into the AVVP task and remove noisy labels on the segment level by constructing prompts and calculating the text-image similarity.

**Language-Supervised Vision Representation Learning.** Language-supervised vision representation learning aims at learning visual representations from language data. Early works for visual representation learning include predicting the bag-of-words representation [37] or training n-gram models [38] on the YFCC100M dataset [39]. While some work like ICMLM [40] and VirTex [41] produce effective visual representations of COCO Captions [42]. Soon after, CLIP [7] quickly draws attention due to its simplicity, scale, and generalization capabilities. CLIP is pre-trained on 400 million image-text pairs using the contrastive learning approach and can be transferred to multi-modal downstream tasks like image caption [43], and video caption [44] due to its zero-shot transfer ability. Thus we employ CLIP as our image encoder to bridge the gap between the image and language descriptions for segment-level label noising.

# 3 Method

In this paper, we propose a novel Language guided Segment-level Label Denoising (LSLD) method for AVVP, which aims to remove the segment-level noise with the help of natural language. The overview of the proposed method is shown in Fig. 1, where language prompts are generated to calculate the similarity with the segments for denoising. Then dynamic re-weighting is adopted to adjust unreliable segment labels. With the newly assigned segment-level label, the network is trained end-to-end to better classify events on segments.

Following, we introduce our method in detail. We begin with the preliminary and our baseline method (Sec. 3.1). Then we illustrate the language prompt construction (Sec. 3.2), segment-level label denoising (Sec. 3.3), and the dynamic re-weighting strategy (Sec. 3.4).

## 3.1 Preliminaries

**Problem Setup and Notations.** Given a $T$-segments audio-visual video sequence $S = \{V_t, A_t\}_{t=1}^T$, $A$ and $V$ are the audio and visual track, respectively, and each segment is of one second long. Our goal is to predict the event label of each segment in audio and visual, which may contain multiple or no events at all. During *evaluation*, for the $t$-th video segment $(V_t, A_t)$, the target is denoted as $(\mathbf{y}_t^a, \mathbf{y}_t^v, \mathbf{y}_t^{av}) \in \mathbb{R}^{1 \times C}$, where $\mathbf{y}_t^a$, $\mathbf{y}_t^v$, and $\mathbf{y}_t^{av}$ are labels for audio, visual, and audio-visual events, respectively, and $C$ is the number of event classes. During *training*, we can only obtain the weak label of the video as a whole, which is not specific to modality and segment.

**Baseline Framework.** To obtain cross-modal and uni-modal information, we adopt HAN [4] as our baseline. HAN [4] extracts visual feature $\mathbf{F}^v = \{\mathbf{f}_t^v\}_{t=1}^T \in \mathbb{R}^{T \times d}$ and audio feature $\mathbf{F}^a = \{\mathbf{f}_t^a\}_{t=1}^T \in \mathbb{R}^{T \times d}$ by Resnet [45] and VGGish [46], where $d$ is the feature dimension. Then it adopts self-attention and cross-attention layers to aggregate cross-modal and uni-modal information at each segment. After gathering the audio and visual features, the segment-level event prediction for audio and visual ($\mathbf{p}_t^a$, $\mathbf{p}_t^v \in \mathbb{R}^{1 \times C}$) can be obtained through an FC layer and a sigmoid function. Then the video-level prediction is aggregated by attentive MMIL pooling, which is formulated as:

$$\mathbf{p}^a = \sum_{t=1}^T \mathbf{w}_t^a \mathbf{p}_t^a, \quad \mathbf{p}^v = \sum_{t=1}^T \mathbf{w}_t^v \mathbf{p}_t^v, \quad \mathbf{p}^{av} = \sum_{t=1}^T \sum_{m=1}^M \mathbf{W}_t[m] \odot \mathbf{P}_t[m], \quad (1)$$

where $\mathbf{W}_t = \{\mathbf{w}_t^a, \mathbf{w}_t^v\}$ denote the temporal attention weights, $\mathbf{P}_t = \{\mathbf{p}_t^a, \mathbf{p}_t^v\}$ includes audio and visual predictions, M is the number of modalities. Finally, binary cross-entropy (BCE) loss is adopted to optimize the model in a weakly-supervised way:

$$\mathcal{L}_{base} = \text{BCE}(\mathbf{y}^{av}, \mathrm{p}^{av}) + \text{BCE}(\overline{\mathbf{y}}^a, \mathbf{p}^a) + \text{BCE}(\overline{\mathbf{y}}^v, \mathbf{p}^v), \quad (2)$$

where $\overline{\mathbf{y}}^a$, $\overline{\mathbf{y}}^v$ are video-level audio and visual labels by performing label smooth on $\mathbf{y}^{av}$ to mitigate the impact of noisy labels. As the baseline method only utilizes video-level labels during training, which is affected by the segment-level noise, we propose a Language guided Segment-level Label Denoising (LSLD) method to address segment-level noise.

## 3.2 Language Prompt Construction

As discussed above, during training, we can only access video-level labels, but for each segment, the label could be any combination of events, which is flexible. To address the flexible label assignment challenge, we introduce the language modality into our task, which describes how the events occur. For each video, to consider all cases of event appearance, we create prompts for each case according to its video-level labels. As shown in Fig. 1 (a), taking a video labeled with two classes named A and B as an example, we construct 4 prompts describing A appears, B appears, both classes appear and none of them appear, respectively. Finally, the language prompts for the video could be:

$$[Class\,A,\ Class\,B,\ Class\,A\,and\,B,\ other],$$

where *other* means no pre-defined event appears.

## 3.3 Language Guided Segment-Level Label Denoising

With language prompts generated in Sec. 3.2, we intend to remove segment-level noisy labels by language guidance. Specifically, we calculate the similarity between each segment and prompts, where the prompt with the highest similarity is more likely to describe the segment accurately. As illustrated in Fig. 1 (b), take the visual modality for example, we feed the constructed language prompts into CLIP's text encoder to obtain the text features, and each frame of the video is fed into CLIP's image encoder to extract the image feature. We denote the text feature as $\mathbf{f}_t \in \mathbb{R}^{N \times d}$ and the visual feature as $\mathbf{f}_v \in \mathbb{R}^{T \times d}$, where $N$ is the number of prompts, $T$ is the number of segments, and $d$ is feature dimension. Then we calculate the text-image similarity matrix $\mathbf{s} \in \mathbb{R}^{N \times T}$ on the normalized text and image feature:

$$\mathbf{s} = \text{Softmax}(\frac{\mathbf{f}_t}{||\mathbf{f}_t||_2} \otimes (\frac{\mathbf{f}_v}{||\mathbf{f}_v||_2})^\top), \quad (3)$$

where $\otimes$ denotes the matrix multiplication. For the audio modality, we feed the audio into CLAP's audio encoder and the language prompts into CLAP's text encoder.

With the similarity matrix, the language prompt with the highest similarity for each segment is obtained, which infers the segments' corresponding event label. For instance, as shown in Fig.1 (b), the video is labeled with class A and class B, while the most similar prompt of its second segment is *Class A*, which means that this segment most likely contains only with class A. In this case, we treat class B as the noisy class of this segment for label denoising. Likewise, if the most similar prompt is *other*, both class A and class B will be treated as noisy labels.

With segment-level label denoising, each audio/visual segment is assigned its own event label $\widetilde{\mathbf{y}}_t^a / \widetilde{\mathbf{y}}_t^v$ instead of being regarded as having the same label as the whole video. In this way, the labels depict the audio/visual segments more accurately which boosts network learning.

### 3.4 Dynamic Re-weighting for Unreliable Segments

In Sec. 3.3, each segment is re-labeled to avoid segment-level noise. However, such noise is inevitable, because some segments may be mislabeled. To further alleviate the impact of mislabeled segments, in this section, we propose to mine the unreliable segments and learn from them with dynamic weight.

To mine the unreliable segments, for each event class, we aggregate the segments with and without the event into two groups (*i.e.*, potentially positive and negative segments) by the denoised segment-level labels. Then, we calculate the similarity between the visual segments and the language prompts of the class for the two groups separately to compare their similarity distribution. The whole process is shown in Fig. 1 (c), where the similarity distributions of with/ without *Car* are presented. We observe that though some visual segments in orange are labeled as without a *Car*, their similarity is still high, while some segments in blue that are labeled as with a *Car* are not that similar to the class description. Therefore, we consider the overlap region of the two similarity groups as mislabeled segments.

Then, we propose a dynamic re-weighting strategy to learn unreliable segments softly. Specifically, for an event and a corresponding unreliable segment, the weighted similarity between the segment and event description is adopted as the soft label. Then, for the t-th segment of a video, its label for event class $c$ can be calculated as:

$$\hat{\mathbf{y}}_t^v[c] = \begin{cases} Min(1, \alpha \times \mathbf{s_c}), & \widetilde{\mathbf{y}}_t^v[c] = 1 \ and \ \mathbf{s_c} \leq Max_{\text{w/o}}, \\ Min(1, \beta \times \mathbf{s_c}), & \widetilde{\mathbf{y}}_t^v[c] = 0 \ and \ \mathbf{s_c} \geq Min_{\text{w}}, \\ \widetilde{\mathbf{y}}_t^v[c], & otherwise, \end{cases} \tag{4}$$

where $Min()$ is a function to return the item with the lowest value, $\alpha$ and $\beta$ are parameters for the segment with/ without the event $c$. $\mathbf{s_c}$ is the similarity between segments and class description using Eq. 3. $Min_{\text{w}}$ is the lowest similarity of the segments with the event, $Max_{\text{w/o}}$ is the highest similarity of segments w/o the event. If the similarity of a segment w/o the event is higher than $Min_w$, then its similarity is set to $Min(1, \beta \times \mathbf{s_c})$. If the similarity of a segment with the event is lower than $Max_{w/o}$, then its similarity is set to $Min(1, \alpha \times \mathbf{s_c})$. Note that, during the application, we set $\alpha > \beta$ to ensure the segments with $\widetilde{\mathbf{y}}_t^v[c] = 1$ still have more possibility of the event appearance. We only adopt dynamic re-weighting on the label of unreliable visual segments.

After dynamic re-weighting, we use the segment-level label to optimize the model on the audio and visual modality. The pipeline is shown in Fig. 1 (d). Specifically, we replace the video-level label $\overline{\mathbf{y}}^v$ with our segment-level label $\hat{\mathbf{y}}_t^v$, and $\overline{\mathbf{y}}^a$ with the segment-level label $\widetilde{\mathbf{y}}_t^a$, in Eq. 2. Notably, we also use CLAP/CLIP to extract the audio/visual features during training since CLAP/CLIP trains from large-scale datasets and learns richer audio/visual information.

## 4 Experiments

### 4.1 Experimental Setup

**Dataset.** In the AVVP task, we only evaluate our method on the Look, Listen and Parse (LLP) Dataset [4] following previous AVVP work. The dataset contains 11,849 video clips 10 seconds long from 25 different event categories, including human activities, animals, musical instruments, *etc*. For the training process, we use 10,000 video clips with only video-level event labels. The remaining

Table 1: **Comparisons with the state-of-the-art methods on LLP dataset.** "CLAP+CLIP" means we use the audio and visual features extracted by CLAP [8] and CLIP [7] respectively, while * represents our baseline framework. The best results are marked in bold.

| Method | Segment-Level | | | | | Event-level | | | | |
|---|---|---|---|---|---|---|---|---|---|---|
| | A | V | A-V | Type | Event | A | V | A-V | Type | Event |
| AVE [1] | 47.2 | 37.1 | 35.4 | 39.9 | 41.6 | 40.4 | 34.7 | 31.6 | 35.5 | 36.5 |
| AVSDN [16] | 47.8 | 52.0 | 37.1 | 45.7 | 50.8 | 34.1 | 46.3 | 26.5 | 35.6 | 37.7 |
| HAN [4]* | 60.1 | 52.9 | 48.9 | 54.0 | 55.4 | 51.3 | 48.9 | 43.0 | 47.7 | 48.0 |
| MA [5] | 60.3 | 60.0 | 55.1 | 58.9 | 57.9 | 53.6 | 56.4 | 49.0 | 53.0 | 50.6 |
| CVCMS [47] | 60.8 | 63.5 | 57.0 | 60.5 | 59.5 | 53.8 | 58.9 | 49.5 | 54.0 | 52.1 |
| DHHN [48] | 61.4 | 63.4 | 56.8 | 60.5 | 59.5 | 54.6 | 60.8 | 51.1 | 55.5 | 53.3 |
| MM-Pyramid [49] | 61.1 | 60.3 | 55.8 | 59.7 | 59.1 | 53.8 | 56.7 | 49.4 | 54.1 | 51.2 |
| MGN [36] | 60.8 | 55.4 | 50.4 | 55.5 | 57.2 | 51.1 | 52.4 | 44.4 | 49.3 | 49.1 |
| JoMoLD [6] | 61.3 | 63.8 | 57.2 | 60.8 | 59.9 | 53.9 | 59.9 | 49.6 | 54.5 | 52.5 |
| CMPAE [50] | 64.2 | 66.4 | 59.2 | 63.3 | 62.8 | 56.6 | 63.7 | 51.8 | 57.4 | 55.7 |
| **LSLD (ours)** | 62.7 | 67.1 | 59.4 | 63.1 | 62.2 | 55.7 | 64.3 | 52.6 | 57.6 | 55.2 |
| **LSLD+CLAP+CLIP** | **68.7** | **71.3** | **63.4** | **67.8** | **68.2** | **61.5** | **67.4** | **55.9** | **61.6** | **60.6** |

Table 2: Ablation study of each component of LSLD. Here, "DeN" denotes language-guided segment-level label denoising. "ReW" denotes the dynamic weighting strategy on unreliable segments. All experiments are based on CLAP and CLIP features.

| DeN | ReW | Segment-Level | | | | | Event-level | | | | |
|---|---|---|---|---|---|---|---|---|---|---|---|
| | | A | V | A-V | Type | Event | A | V | A-V | Type | Event |
| ✗ | ✗ | 64.1 | 54.9 | 49.0 | 56.0 | 60.0 | 53.3 | 51.0 | 41.8 | 48.7 | 51.6 |
| ✓ | ✗ | 67.3 | 70.1 | 61.2 | 66.2 | 67.2 | 60.8 | 66.3 | 54.3 | 60.5 | 60.2 |
| ✓ | ✓ | **68.7** | **71.3** | **63.4** | **67.8** | **68.2** | **61.5** | **67.4** | **55.9** | **61.6** | **60.6** |

1849 validation and test videos possess modality and segment-specific labels (*i.e.*, start and end time of each event on audio and visual track). We conduct experiments following the official data splits from the LLP dataset.

**Evaluation Metrics.** We evaluate the parsing performance of audio, visual, and audio-visual events under both segment-level and event-level metrics by adopting F-scores. The segment-level metrics can evaluate segment-wise event prediction performance. For the event-level metrics, we extract events by concatenating consecutive positive segments in the same event and compute event-level F-score with mIoU=0.5 as the threshold. In addition, we evaluate the parsing performance of audio-visual events using Type@AV and Event@AV metrics. Type@AV is calculated by averaging audio, visual, and audio-visual event evaluation results while Event@AV considers all audio and visual event categories for each sample instead of directly averaging results from different events.

**Implementation Details.** We conduct the training and evaluation processes on a single NVIDIA GTX 2080 Ti GPU with 11 GB memory. Following the data preprocessing in previous works, we decode a 10-second video at 8 fps into 10 segments. Instead of using pre-trained ResNet-152 [45] and R(2+1)D [51], we also utilize CLIP's [7] image encoder (*i.e.*, pre-trained ViT-B/16 [52]) to extract visual features with richer semantic information. Furthermore, we leverage CLAP [8] pretrained on Audioset [53] to extract the audio features instead of using VGGish [46]. Following HAN [4] we adopt the Adam optimizer and the learning rate 2e-4 drops by a factor of 0.25 for every 6 epochs. We train the model with a batch size of 32 for 20 epochs. We also add layers of HAN [4] to increase the complexity of the network for better performance. $\alpha$ is set to 4 and $\beta$ is 0.4.

## 4.2 Comparison with State-of-the-art Methods

To demonstrate the effectiveness of our method, we compare it to several baseline works including the audio-visual event localization approach AVE [1] and AVSDN [16] and the state-of-the-art methods for the weakly-AVVP task, such as HAN [4], MA [5], JoMoLD [6], *etc*.

Table 3: Analysis of different strategies to indicate that no event appears in the prompt.

| Method | Segment-Level | | | | | Event-level | | | | |
|---|---|---|---|---|---|---|---|---|---|---|
| | A | V | A-V | Type | Event | A | V | A-V | Type | Event |
| $[Class, Not Class]$ | 40.8 | 63.1 | 39.3 | 47.7 | 51.9 | 33.8 | 59.8 | 33.5 | 42.3 | 44.4 |
| $[Class, None]$ | 66.3 | **71.4** | 61.2 | 66.3 | 67.1 | 57.9 | **67.8** | 53.5 | 59.7 | 58.7 |
| $[Class, Other]$ | **68.7** | 71.3 | **63.4** | **67.8** | **68.2** | **61.5** | 67.4 | **55.9** | **61.6** | **60.6** |

Table 4: Algorithm analysis of dynamic re-weighting. All results are of visual metric.

(a) Analysis on different $\alpha$ and $\beta$ values.

| $\alpha$ \ $\beta$ | 0.2 | 0.4 | 0.6 | 0.8 |
|---|---|---|---|---|
| 3.5 | 71.1 | 71.1 | 71.3 | 71.0 |
| 4 | 71.2 | **71.3** | 70.9 | 70.6 |
| 4.5 | 70.3 | 70.9 | 70.3 | 70.2 |
| 5 | 69.0 | 70.3 | 70.8 | 69.9 |

(b) Analysis of different re-weighting strategies.

| Method | Visual | |
|---|---|---|
| | Segment-Level | Event-Level |
| re-weight on w/ event | 70.4 | 66.8 |
| re-weight on w/o event | 70.7 | 67.0 |
| stable re-weight | 70.5 | 66.8 |
| dynamic re-weight | **71.3** | **67.8** |

As shown in Table 1, we conduct our experiments on the LLP dataset with all state-of-the-art methods. We observe that our method has competitive performance on all evaluation metrics. Compared with our baseline method HAN [4], the proposed method can significantly improve the performances on visual metrics. Especially, our method gains 14.2 and 10.5 points on segment-level visual parsing and audio-visual parsing metrics with ResNet and VGGish features, respectively. Similar improvements are also observed on event-level metrics.

Compared with state-of-the-art methods, our method still yields better performance on all visual-related metrics. Especially, compared with JoMoLD, we achieved 3.3, 2.2, and 2.3 points of improvement in terms of segment-level visual, audio-visual, and type parsing. With the CLAP and CLIP features, we improve the segment-level audio metrics from 62.7% to 68.7%, and visual metrics from 67.1% to 71.3%.

### 4.3 Ablation Studies

**Effectiveness of language guided segment-level denoising**. The ablation study of denoising is shown in Table 2. We observe that the results on segment-level audio/visual parsing are improved by 3.2%/15.2% when we construct prompts and remove segment-level noisy labels by language guidance. The significant improvement demonstrates that with the help of language description, our method does provide more accurate segment-level labels for both audio and visual modality, which enhances network learning.

**Analysis of different types of prompts.** As shown in Table 3, we investigate the influence of using different language descriptions to express that no event appears in the segment. We observe that when using prompts like $[Class, Not Class]$, the performance is much lower than our method. We suspect the reason is that the text encoder is more sensitive to meaningful words like event class while neglecting the negative word *not*. When using *none*, the result in audio-visual metrics is about 2.2 points lower than ours. The reason is that though the pre-defined events do not appear, background scenes like sky and trees may exist. When using the description *other*, we obtain the best performance, where we neither throw away the possibility of the appearance of background classes nor introduce event categories that do not appear.

**Effectiveness of dynamic re-weighting.** The ablation study of dynamic re-weighting is shown in Table 2. We observe that the results on visual parsing are improved by 1.2% and 1.1% on segment-level and event-level, respectively. The improvement demonstrates that our method accurately finds the possible mislabeled segments to learn them softly.

**Analysis of different $\alpha$ and $\beta$ values.** According to Eq. 4, the hyper-parameter $\alpha$ and $\beta$ are used to be multiplied with similarity to serve as the weight for unreliable segments with/ without the event, respectively. In Table 4 (a), as $\alpha$ increased from 3.5 to 4, the results are usually improved, as our denoising method identifies most of the segments into the correct category, so a larger weight should

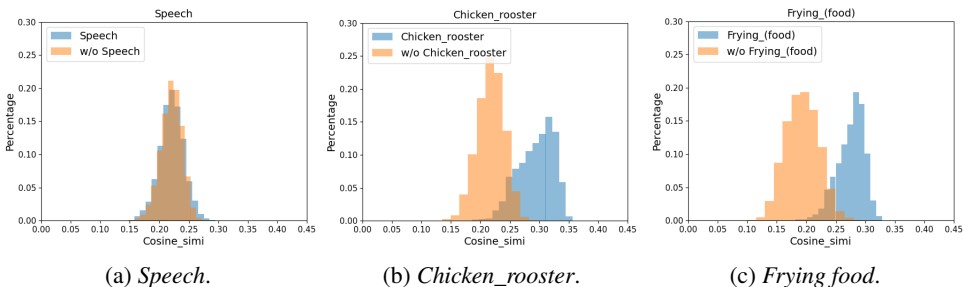

| (a) *Speech*. | (b) *Chicken_rooster*. | (c) *Frying food*. |

Figure 2: Similarity percentage distribution of the visual modality. The similarity is calculated between the visual segments and the event description. We split the segments into with (blue)/ without (orange) the event. Examples of three events are shown.

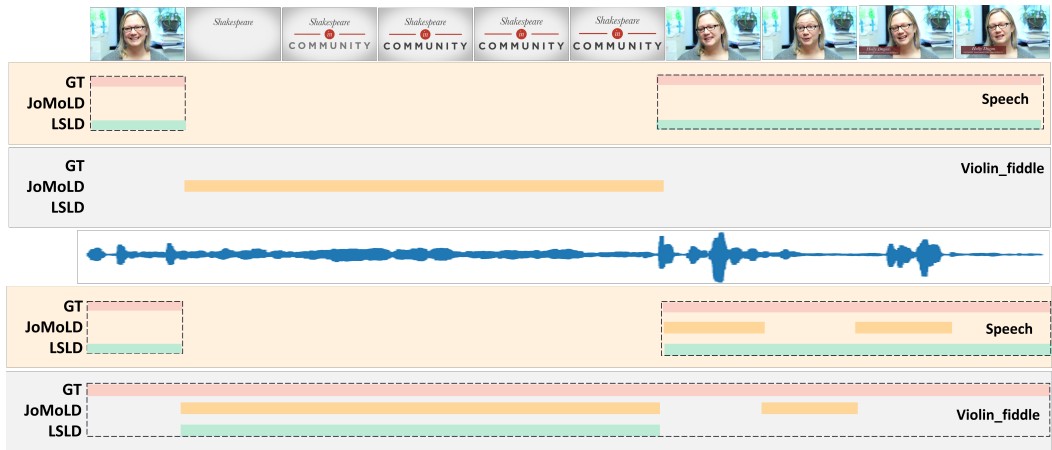

Figure 3: Qualitative result on the LLP test set. The results of our method (LSLD) and the state-of-the-art method JoMoLD [6] are shown. "GT" denotes the ground truth annotations.

be given. However, when $\alpha$ is too big, the performance drops, because most of the labels will remain at 1 without any re-weighting to unreliable segments. A similar observation is found in $\beta$.

**Analysis of different re-weighting strategies.** As shown in Table 4 (b), four different strategies are compared. The first two rows denote conducting dynamic re-weighting on only segments with or without the event, where the results are 0.9% and 0.6% lower than ours on segment-level visual metrics, respectively. "stable re-weight" denotes using stable value instead of weighted similarity as the soft label, where we find the results are 0.8% and 1% lower than ours in terms of segment-level and event-level. This demonstrates that since segments with higher similarity are more likely to belong to the category, re-weighting with dynamic similarity helps to better learn the unreliable segments.

In addition, we visualize three examples of similarity distribution with/ without an event in Fig. 2. We observe that in most cases, such as *chicken rooster* and *frying food*, the similarity distribution is well separated, while only a small area is taken as unreliable segments. However, there is a special case, *speech*, where we can hardly recognize whether it happens through the similarity with the language prompt of the class. We suspect the reason is that *speech* is difficult to recognize in vision. For example, the vision is about a basketball game while *speech* could only occur in the audio track.

### 4.4 Qualitative Results

**Visualization of video parsing results.** We compare our method with the SOTA method JoMoLD [6] by visualizing the audio and visual parsing results in Fig. 3. As for the visual parsing task, the visual track only contains *speech* on the segments, where JoMoLD recognizes it as the *Violin_fiddle*, and our method identifies the *speech* event and localizes the temporal boundaries correctly. The results prove

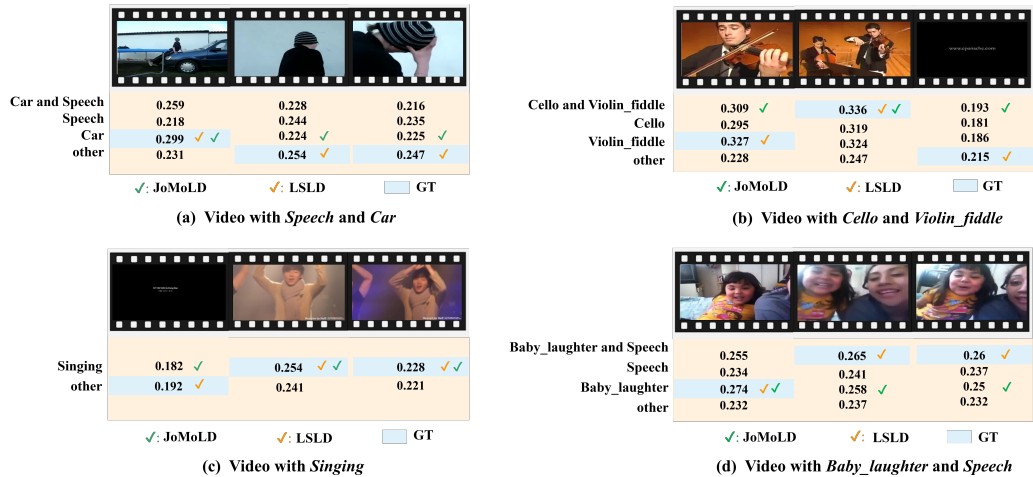

Figure 4: Visualization of segment-level label denoising on visual track. The similarities between our prompts and segments are presented. Results on JoMoLD [6] and our method are shown. Each example presents three segments of a video.

that our approach is more effective in recognizing events in visual segments. As for the audio parsing task, our LSLD properly identifies the segments with *speech*, yet JoMoLD only recognizes *speech* on some of the segments. This demonstrates that LSLD not only works well on the visual track but is also capable of identifying the events that appear on the audio segment as well. However, when identifying the *Violin_fiddle*, both LSLD and JoMoLD only recognize the event in some segments due to the interference of the speech voice, which impels us to further refine the method on the audio modality.

**Visualization of the denoising procedure.** We visualize the segment-level label denoising results of our method and JoMoLD [6] in Fig. 4. We observe that, in all examples, our prompt of the highest similarity captures the appeared events accurately. As shown in Fig. 4 (a), in the last two segments there is no event exists, where JoMoLD still recognizes *Car* because *Car* appears in the first segment and JoMoLD can only denoise at video-level. In contrast, LSLD correctly assigns labels for all the segments. In Fig. 4 (b), LSLD recognizes different event appearances on the segment level, while JoMoLD fails to do so, which demonstrates the importance of segment-specific label assignment and the effectiveness of our method. In Fig. 4 (c), LSLD successfully removes *Singing* on the first segment, yet JoMoLD is unable to remove it. A similar observation could be found in Fig. 4 (d).

## 5 Conclusion and Discussion

In this paper, we address the weakly-supervised audio-visual video parsing task. Instead of focusing on the overall video instance level denoising, we propose to assign specific segment-level labels for each video on the audio and visual track. Since each segment label is not predefined but could be any combination of events that occur in the video, we introduce the language modality into our task and construct language prompts to represent all cases of event appearance. The similarities between each segment and prompts are calculated, where the prompt with the highest similarity is considered the segment-level label. To further alleviate the effect of segment-level label noise, we apply dynamic weighting on visual segments that may be mislabeled. Extensive experiments and qualitative results show that our method outperforms the state-of-the-art methods on the visual metrics by a large margin.

**Limitation.** Since only the LLP dataset currently has fine-grained (segment-level and modality-level) annotations and labels for audio-visual understanding, we only perform LSLD on the LLP dataset. In future work, more effort will be made to explore the effectiveness of our method on other audio-visual datasets.

**Broader impact.** Audio-Visual Video Parsing (AVVP) has numerous potential real-life applications, such as professional media production, audiovisual archive management, entertainment, etc. Our

method provides a new perspective to address AVVP, *i.e.*, through the guidance of natural language, which could benefit from large-scale vision-language models and will inspire future works.

**Acknowledgment.** This work was supported in part by the National Natural Science Foundation of China under Grant 62001331, in part by the Natural Science Foundation of Hubei Province under Grant 2021CFB475, and in part by the Special Fund of Hubei Luojia Laboratory under Grant 220100018.

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
