# Supplementary Material for
# Revisit Weakly-Supervised Audio-Visual Video Parsing from the Language Perspective

**Yingying Fan, Yu Wu, Bo Du, Yutian Lin**[*]
School of Computer Science, Hubei Luojia Laboratory, Wuhan University
{fanyingying_cs, wuyucs, dubo, yutian.lin}@whu.edu.cn

## Appendix

In the supplementary material, we first conduct more ablation studies for our method in Sec. A. In Sec. B, we provide more examples of the similarity distribution with/without the event and visualize more label denoising cases to validate the effectiveness of our method.

## A   More Ablation Studies

In this section, we conduct more ablation studies on the proposed mechanism, including performing LSLD on different SOTA methods, studying the impact of changing class names to make the constructed prompt more contextual and the effectiveness of denoising labels both on audio and visual tracks

**Effectiveness with different SOTA methods.** To investigate the flexibility of our approach, we combine LSLD with different SOTA methods for the AVVP task. As illustrated in Table 1, the results of all methods are improved on both audio and visual metrics with LSLD, especially on the segment-level visual metrics, which is 13 points higher than MM-Pyramid [1] when combined with our approach. The experiments show that our denoised labels are indeed influential and can be properly employed on different SOTA methods.

Table 1: Apply our method to different state-of-the-art methods for the AVVP task. All experiments are conducted with ResNet [2] and VGGish [3] features.

| Method | Segment-Level | | | | | Event-level | | | | |
|---|---|---|---|---|---|---|---|---|---|---|
| | A | V | A-V | Type | Event | A | V | A-V | Type | Event |
| MM-Pyramid [1] | 60.9 | 54.4 | 50.0 | 55.1 | 57.6 | 52.7 | 51.8 | 44.4 | 49.9 | 50.5 |
| MM-Pyramid + LSLD | **61.8** | **67.4** | **60.8** | **63.3** | **61.4** | **54.8** | **63.4** | **54.3** | **57.5** | **53.8** |
| MA [4] | 60.3 | 60.0 | 55.1 | 58.9 | 57.9 | 53.6 | 56.4 | 49.0 | 53.0 | 50.6 |
| MA + LSLD | **61.2** | **66.6** | **59.0** | **62.3** | **61.1** | **54.3** | **64.0** | **52.5** | **56.9** | **54.2** |
| JoMoLD [5] | 61.3 | 63.8 | 57.2 | 60.8 | 59.9 | 53.9 | 59.9 | 49.6 | 54.5 | 52.5 |
| JoMoLD + LSLD | **62.3** | **66.1** | **58.7** | **62.4** | **61.8** | **55.8** | **63.6** | **52.2** | **57.2** | **55.0** |

**Effectiveness of modifying class names in prompts.** In the implementation, we try to modify the prompts to make them more natural instead of using pure event class names. From the results in Table 2, we can see that the segment-level visual metric improves by 1.7 points when we add *playing*

---

[*]Corresponding author

37th Conference on Neural Information Processing Systems (NeurIPS 2023).

before the class names related to musical instruments and replace *basketball bounce* with *playing basketball*. As we transform objects like *Accordion* into human behavior (i.e. *playing the Accordion*), the prompt becomes more natural when they are connected with events related to human actions. For instance, we change *Singing and Accordion* into *Singing and playing the Accordion*. Furthermore, our approach is much better than the one that adds *a photo of* before the prompt since the result of the segment level is 3.2 points higher. While adding *a person* before the class related to human activity would make the prompt more natural, for example, *a person speeching and playing the guitar*, *a person* would introduce a new category (i.e. person) to the sentence, so it is better to add only action-related words as in our method.

Table 2: Study the impact of varying class names to make the prompt more contextual. only visual metrics are reported for more explicit comparisons. All experiments are conducted with CLIP [6] and CLAP [7] features.

| Class Description | Visual | |
|---|---|---|
| | Segment-Level | Event-Level |
| Original Class Desc | 69.6 | 65.7 |
| Desc + playing | **71.3** | **67.4** |
| Desc + a photo of | 68.1 | 64.1 |
| Desc + playing + a person | 69.4 | 66.0 |

Table 3: Study the effectiveness of denoising labels both on audio and visual tracks.

| Method | Segment-Level | | | | | Event-level | | | | |
|---|---|---|---|---|---|---|---|---|---|---|
| | A | V | A-V | Type | Event | A | V | A-V | Type | Event |
| audio only | 66.2 | 54.8 | 52.5 | 57.8 | 59.7 | 58.7 | 50.6 | 45.8 | 51.7 | 54.9 |
| visual only | 64.0 | 70.1 | 61.5 | 65.2 | 63.8 | 53.2 | 66.4 | 53.1 | 57.6 | 53.3 |
| audio and visual | **68.7** | **71.3** | **63.4** | **67.8** | **68.2** | **61.5** | **67.4** | **55.9** | **61.6** | **60.6** |

**Effectiveness of denoising labels both on audio and visual tracks.** In Table 3, we can clearly observe that when we only apply label denoise on the audio or visual track, the capability of the model to recognize events drops significantly. When we only denoise the audio labels, the visual metrics at the segment level drop by 16.5%, whereas when we denoise only the visual labels, the segment-level audio metrics fall by 4.7%, which proves the effectiveness of denoising both the audio and visual labels simultaneously.

## B   Additional Qualitative Analyses

In this section, we visualize more cases of similarity percentage distribution with or without the event and perform more examples of the label denoising procedure.

**Visualization of similarity percentage distribution.** As shown in Figure 1, we can observe that the intersection area on sound-related events is relatively large. We argue that identifying events like *Singing* is more difficult as it requires observation of human facial micro-expressions. Conversely, the model will be more accurate when it comes to recognizing obvious objects or animals like *basketball*, *fire alarm*, *Cat*, *Blender*, *etc*. Thus, we need to perform dynamic re-weighting on the segments with unreliable labels, especially for sound-related events.

**Visualization of segment-level label denoising cases.** In Figure 2 and Figure 3, we present several visualization results of the label denoising process. For a more comprehensive analysis of the effectiveness of our approach, we visualize the label assignment of various event categories, such as *Fire alarm*, *Helicopter*, *basketball bounce*, *etc*. We can observe in Figure 2 (a), JoMoLD [5] removes *speech* on the video-level label while *speech* appears in several segments, the same case happens in Figure 2 (g). In Figure 2 (b), there is no one *Singing* in the first 2 segments, yet JoMoLD still recognizes *Singing* since it can only denoise at video-level, and LSLD correctly assigns the right label for the segments. A similar observation can be found in Figure 2 (c), (d), (e). Note that, we cannot assign the *Car* label to the segments in Figure 2 (c) because "GT" does not contain it. As

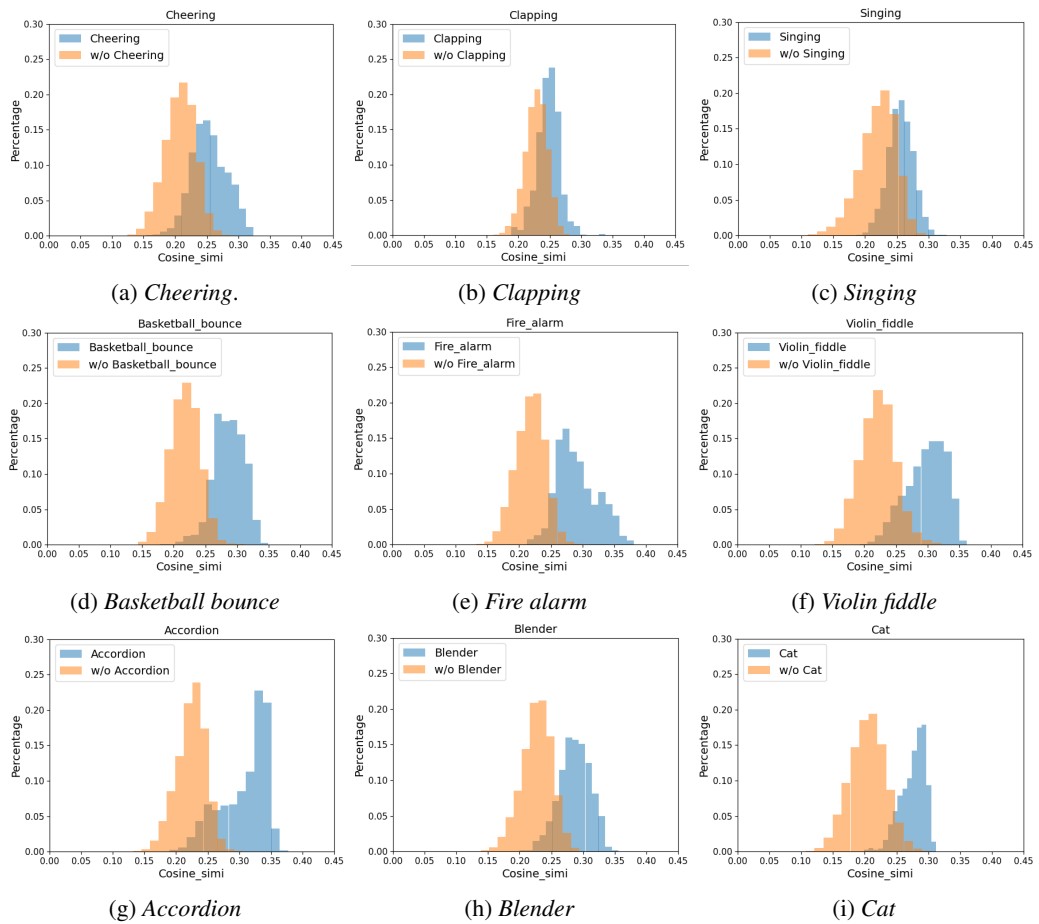

Figure 1: Similarity percentage distribution. The similarity is calculated between the visual segments and the event description. We split the segments into with (blue)/ without (yellow) the event.

shown in Figure 2 (f), LSLD recognizes *Speech* and only removes *Clapping* for the first 2 segments, while JoMoLD fails to do so. Likewise, from Figure 2 (h), we can clearly see that *Dog* is not present in the first segment and *Cat* cannot be seen in the last one, but JoMoLD recognizes both *Cat* and *Dog* in all three of them. In conclusion, LSLD provides better label denoising capability than JoMoLD.

We provide more visualization cases for events like *Chicken rooster*, *Blender*, *Chainsaw* in Figure 3. LSLD successfully removes *Chicken rooster* in Figure 3 (a) when there is no chicken around. Similar cases are found in Figure 3 (c), (d), (e), (g). In addition, JoMoLD again fails to recognize *Speech* in some segments as illustrated in Figure 3 (b), which indicates that JoMoLD is more likely to recognize obvious events while our method is better at discriminating more classes. In Figure 3 (f), a person is *Singing* in the first segment with no one playing the *Accordion*, and LSLD removes *Accordion* from the segment label correctly, which indicates the effectiveness of our method. Also, in the last case of Figure 3, though JoMoLD performs label denoising accurately on the video level, *Motorcycle* only appears in the last segment. On the other hand, LSLD provides a reasonable denoising of the labels at the segment level.

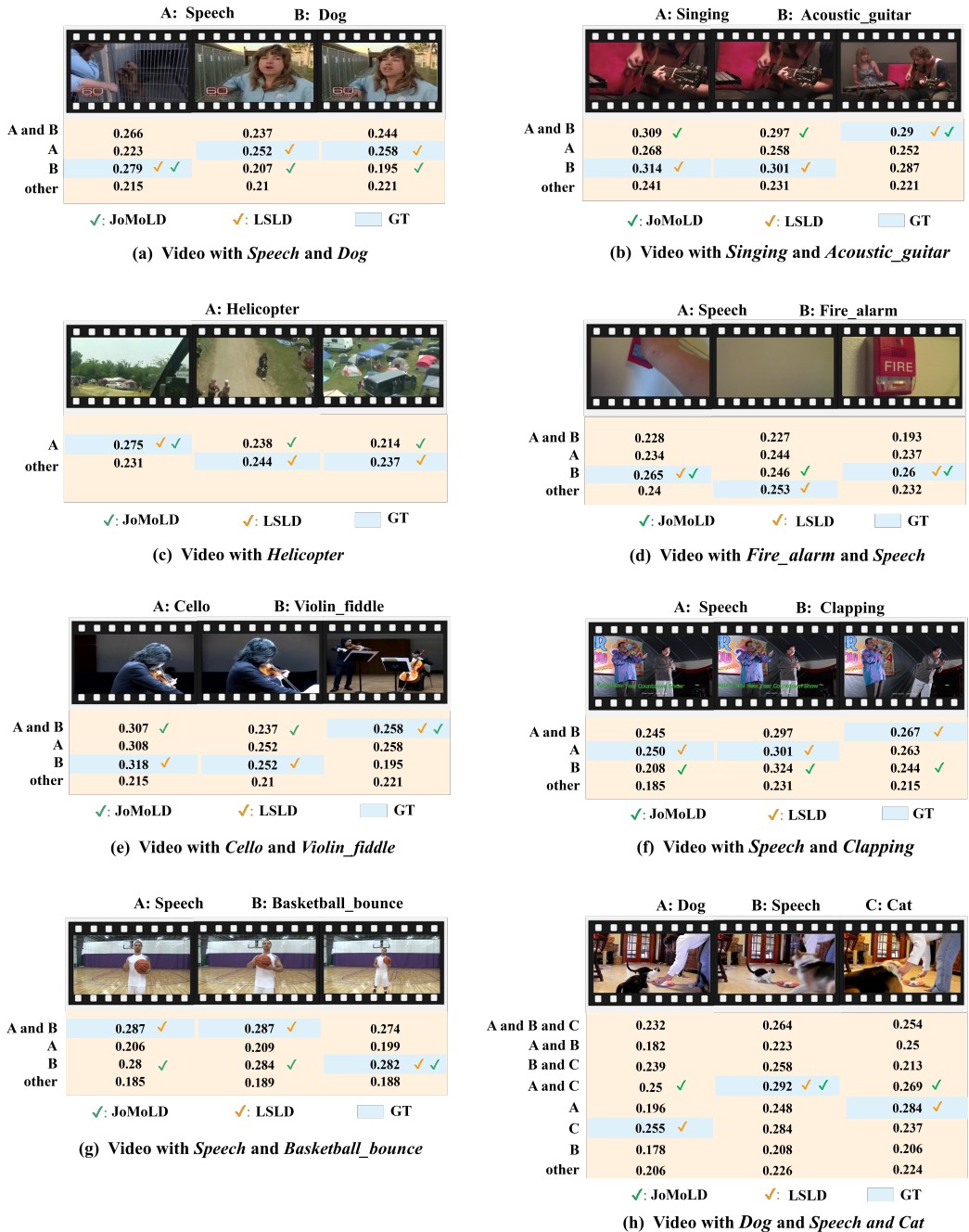

Figure 2: Visualization of segment-level label denoising on visual track. The similarity between our prompts and segments are presented. Results on JoMoLD and our method are shown. Each example presents three segments of a video. "GT" denotes the ground truth.

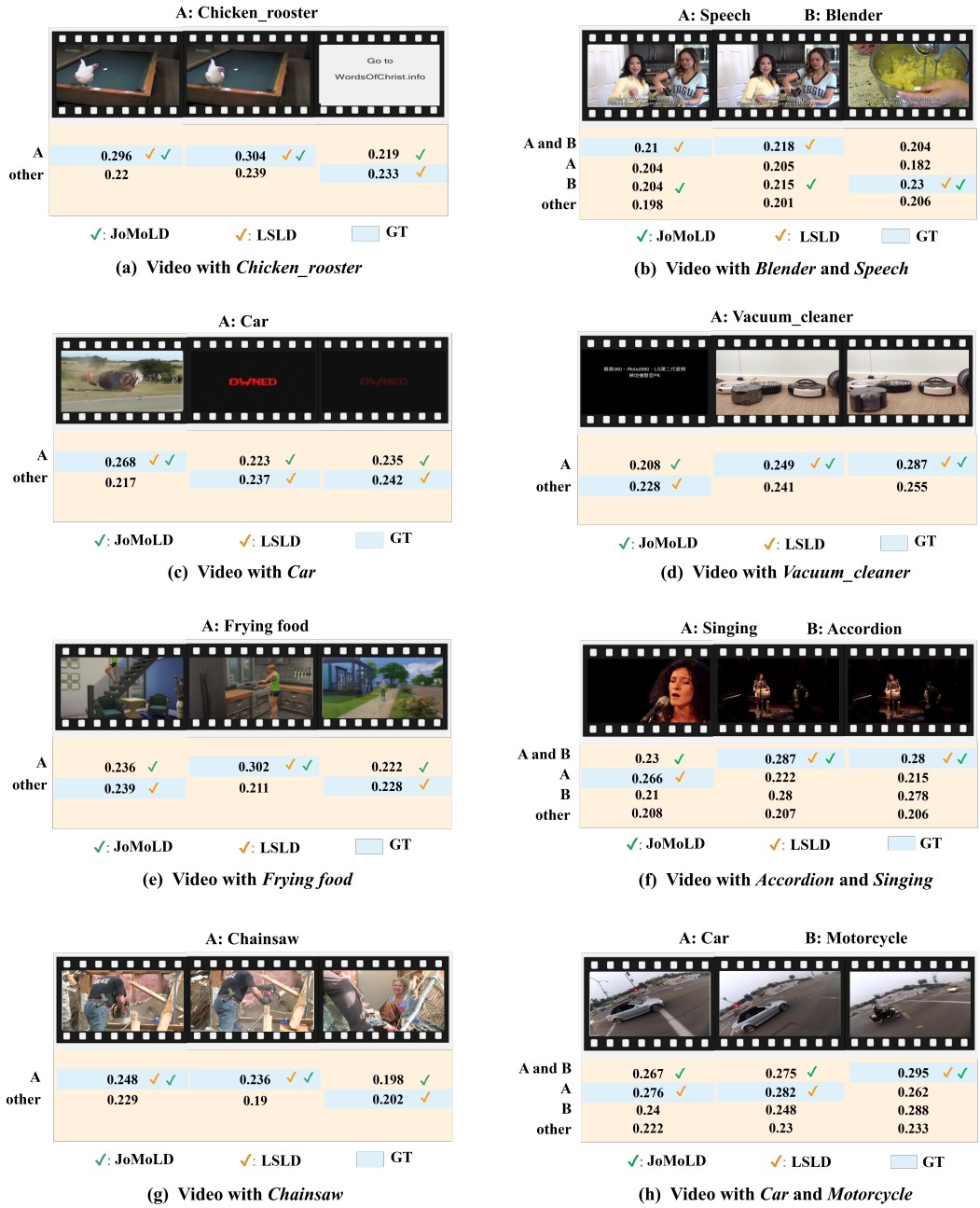

Figure 3: Visualization of more segment-level label denoising cases.