# OpenReview forum: "Revisit Weakly-Supervised Audio-Visual Video Parsing from the Language Perspective"
_NeurIPS.cc/2023/Conference — NeurIPS 2023 poster_

### Official Review · Reviewer_2DCt · 2023-07-05

**Soundness:** 3 good
**Presentation:** 3 good
**Contribution:** 3 good
**Rating:** 7
**Confidence:** 3

**Summary:**

The goal of this paper is to create a model that can solve the task of audio-visual video parsing in a weakly-supervised way. For this, the authors propose to use the language modality which proves to provide benefits and improve the performance of the proposed model over the previous baseline. The language modality is used to provide additional supervision for the system. The paper provides an extensive comparison with previous baselines. Also, the authors provide extensive ablations showing the importance of each component and some qualitative examples which provide another perspective of how the model behaves.

**Strengths:**

- The proposed system performs better than the previous baselines.
- Multiple baselines were used to compare against the proposed method.
- Extensive ablation showing the importance of each component
- Presence of qualitative results.


**Weaknesses:**

- Line 2: typo “andvisual”->”and visual”
- Figure 1: last line of the description: the cross product looks different than in the figure. Make them consistent.
- Line 127: Do the authors also train the Resnnet and VGGish? Or are they pre-trained and then frozen?
- Equation 1. I have some questions here. What is the shape of w^a_t and w^v_t? Are these scalars or vectors? Moreover, p_t^a/p_t^v is a vector. What will p^a and p^v? Also vectors? If w is a scalar then p^a/p^v will be vectors. If w is a vector, then I would assume there is a dot product between w and p which may be wrong depending on the dimension of w. Is there a dot product in the p^a and p^v? Moreover, if there is a dot product then p^a and p^v will be a single scalar, and not a tensor. However, I think the authors want to obtain p^a which will have the same shape as p^a_t, but it will aggregate the scores from the whole video. I also observed that p^{av} uses a Hadamard product. The first thing would be to clarify the shape of the weights and that would also clarify a lot of things in the equations. Right now, for me, it is very hard to understand what is going on there.
- Line 154: How are the features per segment obtained? CLIP gives features per frame. Do the authors average the per-frame features from CLIP to obtain per-segment features?
- Line 179: How is the overlap region chosen? What are the thresholds? It is said before that “...some segments in orange are labeled as without car, their similarity is still high…”. How high and in comparison with what? Thus, provide some details on how it is judged if the similarity is still too high or too low.
- Line 182: is this t-th segment one of the unreliable segments, or it is just the t-th segment of the video? Make this explicit, as right now it seems that it is just the t-th segment of a video, but I understood that the re-weighting is only applied to the unreliable segments, as mentioned in Line 170.
- Line 184: What are the values of alpha and beta?
- Line 191-194. I am a bit confused by these lines. First, I will refer to the figure. In the upper part of Figure 1 (d), it is seen that the method is learning in a supervised way to label segments correctly, based on the labels that are provided by the denoising and the re-weighting mechanism. However, for the lower part of Figure 1 (d), only the VGGish features are used for the MMIL. Is this correct? Shouldn’t both the visual and the audio features be used by MMIL, as in equation (1)? As far as I understand the only thing that changes is the loss of the video modality, which now can provide segment-level supervision. But everything else is staying the same.
- Line 193: Here it is said that CLIP is used to extract the visual features during training. Does this mean that the authors train HAN but they replace the Resnet with CLIP? If yes, is CLIP trained, or it is frozen?
- Line 230: typo “leave the label” -> “leave out the label”


**Questions:**

Most of the above questions can be easily fixed and clarified. For now, I think this paper looks good, so my score is weak accept. However, I would encourage the authors to clarify the mathematical equations, as this will make the proposed method easier to be understood.

**Limitations:**

The authors discuss the limitations of their work and how the work can be extended.

---

> ### Author Rebuttal · Authors · 2023-08-10
>
> We thank you for your reviews and address your concerns as follows.
>
> **Q1**: Do the authors also train the Resnet and VGGish? Or are they pre-trained and then frozen?
>
> **A1**: Yes, they are pre-trained and then frozen. The Resnet is pre-trained on ImageNet and VGGish is pre-trained on Audioset.
>
> **Q2**: What is the shape of $\mathbf{w}^a_t$ and $\mathbf{w}^v_t$? Are these scalars or vectors? Moreover, $\mathbf{p}_t^a$ /$\mathbf{p}_t^v$ is a vector. What is $\mathbf{p}^a$ and $\mathbf{p}^v$? Also vectors? Is there a dot product in the $\mathbf{p}^a$ and $\mathbf{p}^v$? Moreover, if there is a dot product then $\mathbf{p}^a$ and $\mathbf{p}^v$ will be a single scalar and not a tensor.
>
> **A2**: We apologize for not describing this information clearly in the paper. $\mathbf{w}_t^a,\mathbf{w}_t^v\in\mathbb{R}^{1\times{C}}$ are all vectors, whose shape is the same as $\mathbf{p}_t^a$ and $\mathbf{p}_t^v$. $\mathbf{p}^a$, $\mathbf{p}^v$ are the weighted average of $\mathbf{p}_t^a$, $\mathbf{p}_t^v$ respectively. So $\mathbf{p}^a$, $\mathbf{p}^v$ are also vectors. We will describe these weights and equations more explicitly in the paper.
>
> **Q3**: How are the features per segment obtained? CLIP gives features per frame. Do the authors average the per-frame features from CLIP to obtain per-segment features?
>
> **A3**: Yes, since a segment/second is composed of 8 frames, we average the features of the 8 frames to get the feature of a segment.
>
> **Q4**: How is the overlap region chosen? What are the thresholds? It is said before that “...some segments in orange are labeled as without car, their similarity is still high…”. How high and in comparison with what? Thus, provide some details on how it is judged if the similarity is still too high or too low.
>
> **A4**: We apologize that we didn't describe it clearly. For positive samples ($\mathbf{\widetilde{y}}_t^v[c] = 1$), the unreliable one is that its prompt similarity score $\mathbf{s}_c$ is lower than the maximum similarity scores of negative data ($Max_w/o$). For negative samples, the unreliable one is that its prompt similarity score $\mathbf{s}_c$ is higher than the minimum similarity scores of positive data ($Min_w$). Indeed the unreliable data are the intersection of the yellow region (negative data) and blue region (positive data) as shown in Fig. 2 of the original manuscript. We then change the labels of unreliable samples by a soft value (proportional to its prompt similarity), rather than the hard values (0 or 1). For those reliable ones, we keep their original label unchanged. Thus the detailed procedure for applying dynamic re-weighting on unreliable segments is illustrated below.
> $$\mathbf{\hat{y}}_t^v[c]=
> \begin{cases}
> Min(1,\alpha \times \mathbf{s_c} ) & ,\mathbf{\widetilde{y}}_t^v[c] = 1 \\ and \\  \mathbf{s_c} \leq Max_w/o \\\\
> Min(1,\beta \times \mathbf{s_c})&, \mathbf{\widetilde{y}}_t^v[c] = 0 \\ and \\ \mathbf{s_c} \geq Min_w\\\\
> \widetilde{y}_t^v[c]& , otherwise,
> \end{cases}$$
> where $ \mathbf{s_c}$ is the similarity between the segment and the event, $Min_w$ is the lowest similarity of the segments with the event, $Max_w/o$ is the highest similarity of segments w/o the event, $\alpha$ and $\beta$ are two scalars.
>
> **Q5**: Is this t-th segment one of the unreliable segments, or it is just the t-th segment of the video? Make this explicit, as right now it seems that it is just the t-th segment of a video, but I understood that the re-weighting is only applied to the unreliable segments, as mentioned in Line 170.
>
> **A5**: Thanks for pointing out! In the original Eq. (4), t-th segment is one of the unreliable segments. The Eq.(4) will be updated as replied in the question above (Q4).
>
> **Q6**: What are the values of alpha and beta?
>
> **A6**: We set $\alpha$ as 4 and $\beta$ as 0.4, because the similarity of segments with the event is supposed to be higher than segments w/o the event. We'll add the value of these two parameters in the implementation details. Besides, the analysis of different $\alpha$ and $\beta$ values is shown in Table 4 (a) of the original manuscript.
>
> **Q7**: I am a bit confused by these lines. First, I will refer to the figure. In the upper part of Figure 1 (d), it is seen that the method is learning in a supervised way to label segments correctly, based on the labels that are provided by the denoising and the re-weighting mechanism. However, for the lower part of Figure 1 (d), only the VGGish features are used for the MMIL. Is this correct? Shouldn’t both the visual and the audio features be used by MMIL, as in equation (1)? As far as I understand the only thing that changes is the loss of the video modality, which now can provide segment-level supervision. But everything else is staying the same.
>
> **A7**: Yes, only the VGGish features (i.e. audio features) are used for the MMIL. The reason of using MMIL is to aggregate the segment-level predictions to the video-level features predictions for weakly supervised optimization. Since we generated the segment-level labels for the visual modality, we have already turned the weakly supervised labels into the fully supervised labels.  Thereby we do not need to perform the MMIL mechanism again. Sorry we didn't make it clear in Figure 1, actually we use the BCE (binary cross-entropy) loss for the video modality, and the final loss function is:
> \begin{equation}
>     \mathcal{L} = \mathrm{BCE}(\mathbf{y}^{av}, \mathrm{p}^{av}) + \frac{1}{T}\sum_{t=1}^{T}\mathrm{BCE}(\mathbf{\hat{y}}_t^v, \mathbf{p}^v_t) + \mathrm{BCE}(\overline{\mathbf{y}}^a, \mathbf{p}^a),
> \end{equation}
>
> **Q8**: Here it is said that CLIP is used to extract the visual features during training. Does this mean that the authors train HAN but they replace the Resnet with CLIP? If yes, is CLIP trained, or it is frozen?
>
> **A8**: Yes, we replace Resnet with CLIP Image Encoder. And CLIP is frozen during training.

---

> > ### Comment · Reviewer_2DCt · 2023-08-11
> >
> > I thank the authors for addressing my concerns in their rebuttal. I think the paper is technically strong and it provides interesting results. However, one of the reviewers pointed out that the evaluation is done only on one dataset, LLP. This is not ideal and a comparison on multiple other datasets would have made the paper stronger. Thus, I will not increase my score (even if all my concerns were addressed). However, I will also not decrease my score because of this drawback, as I still think that even with one single dataset, the paper provides some interesting insights that could benefit the research community.

---

> > > ### Author Response · Authors · 2023-08-12
> > > **Thanks for acknowledging the technical strength and interesting insight of our paper!**
> > >
> > > We greatly appreciate your acknowledgment of the technical strength and interesting insight of our paper!
> > >
> > > Unfortunately, only the LLP dataset currently has fine-grained (segment-level and modality-level) annotations and labels for audio-visual understanding. That's why state-of-the-art works (e.g., HAN [1], MA [2], CVCMS [3], MGN [4], JoMoLD [5], DHHN [6], MM-Pyramid [7], CMPAE [8]) and our work solely tested their models on a single dataset.
> > >
> > > In the future, we will try to collect and annotate more new fine-grained audio-visual datasets to facilitate the research community.
> > >
> > > [1] Tian et al., Unified multisensory perception: Weakly-supervised audio-visual video parsing. In ECCV, 2020.
> > >
> > > [2] Wu et al., Exploring heterogeneous clues for weakly-supervised audio-visual video parsing. In CVPR, 2021.
> > >
> > > [3] Lin et al., Exploring cross-video and cross-modality signals for weakly-supervised audio-visual video parsing. In NeurIPS, 2021.
> > >
> > > [4] Mo et al., Multi-modal grouping network for weakly-supervised audio-visual video parsing. In NeurIPS, 2022.
> > >
> > > [5] Cheng et al., Joint-modal label denoising for weakly-supervised audio-visual video parsing. In ECCV, 2022.
> > >
> > > [6] Jiang et al., Dhhn: Dual hierarchical hybrid network for weakly-supervised audio-visual video parsing. In ACM MM, 2022.
> > >
> > > [7] Yu et al., Mm-pyramid: Multimodal pyramid attentional network for audio-visual event localization and video parsing. In ACM MM, 2022.
> > >
> > > [8] Gao et al., Collecting Cross-Modal Presence-Absence Evidence for Weakly-Supervised Audio-Visual Event Perception. In CVPR, 2023.

---

> > > > ### Comment · Reviewer_2DCt · 2023-08-13
> > > >
> > > > I thank the authors for providing more details. At a closer inspection it can be seen that those baselines use the same dataset.
> > > >
> > > > However, from the baseline list, the last three are missing in the comparison table in the paper. Why? They are the most recent ones and they are ignored. Especially CMPAE, which is the most recent one and which in some metrics is actually better than the proposed method. I think putting more recent baselins (and removing the old ones if the space is an issue) would be better.

---

> > > > > ### Author Response · Authors · 2023-08-13
> > > > > **Author's Response**
> > > > >
> > > > > For DHHN [1] and MM-Pyramid [2], we did not list them in the manuscript due to space limits. However, we did compare our method with a better state-of-the-art method JoMoLD [3], whose performance is higher than DHHN and MM-Pyramid. In addition, we did combine DHHN and MM-Pyramid with our method respectively in the appendix (see Table 1 of the original supplementary pdf file) and show significant improvement over these two methods.
> > > > >
> > > > > For CMPAE [4], which was published in CVPR 23, we did not compare it with our method in the original submissions since it came out after our submission. The paper was first online in June 2023, but the NeurIPS 23 submission deadline was May 2023.  Besides the authors of CMPAE did not put their paper in Arxiv in advance. We will include this paper in our final version.
> > > > >
> > > > > [1] Jiang et al., Dhhn: Dual hierarchical hybrid network for weakly-supervised audio-visual video parsing. In ACM MM, 2022.
> > > > >
> > > > > [2] Yu et al., Mm-pyramid: Multimodal pyramid attentional network for audio-visual event localization and video parsing. In ACM MM, 2022.
> > > > >
> > > > > [3] Cheng et al., Joint-modal label denoising for weakly-supervised audio-visual video parsing. In ECCV, 2022.
> > > > >
> > > > > [4] Gao et al., Collecting Cross-Modal Presence-Absence Evidence for Weakly-Supervised Audio-Visual Event Perception. In CVPR, 2023.

---

> > > > > > ### Comment · Reviewer_2DCt · 2023-08-14
> > > > > >
> > > > > > Thank you for your response. The authors addressed my concerns. As a result, I will increase my score to Accept.

---

> > > > > > > ### Author Response · Authors · 2023-08-15
> > > > > > > **Thanks for recognizing the paper!**
> > > > > > >
> > > > > > > Thanks for recognizing our work and increasing the score! We will take all the suggestions into the revised manuscript.

---

### Official Review · Reviewer_Nz2w · 2023-07-06

**Soundness:** 3 good
**Presentation:** 3 good
**Contribution:** 3 good
**Rating:** 5
**Confidence:** 4

**Summary:**

This paper tackles the audio-visual parsing task by dividing the video-level label into segment-level labels with the help of language based on CLIP. Training with the fine-grained segment-level labels, instead of the coarse video-level labels, makes the model perform better. In the process, the processing for noisy and unreliable labels bring obvious improvement on the task.

**Strengths:**

The presentation is clear and it’s easy to follow.
The manner to introduce CLIP to generate segment-level labels to replace video-level labels for training is smart, simple and effective.


**Weaknesses:**

I think the main contribution in this paper is to introduce large models such as CLIP to help the audio-visual video parsing task, and the exploration could be more thorough. The main task is the “audio-visual” video parsing task, then how would the relationship between the two modalities influence the task?
In default case, would the audio modality perform better than visual modality? In Table 1,2 and so on, the results show that “A” always perform better than “V” when V has not been helped by CLIP; and Even when introducing CLIP, the “HAN +CLIP” also shows that “A” better than “V”. Does this mean that “A” always perform better than “V” by default? And also compared the results of “A-V” and others with “A”, the improvements are small. If so, enhancing the audio modality may be more effective for this task, while the audio modality in this work is weaken in the process. This is because that there is no proper audio-based large models like CLIP?


**Questions:**

How to generate the prompts? how to obtain A and B? Randomly choose a negative class from 25 categories together with the original class to compose A and B respectively? And the paper says that “the language prompts of the video [could] be xx”. Are there any other formats besides the “other/none/notclass” format in experiment?

**Limitations:**

One main motivation of this paper it to tackle the problem of unmatching between the audio and visual modality. Then what’s the actual situation in the evaluated data? And it would be better to provide a statistics of this unmatching in this data to show that the data is proper to evaluate this point, together with quantitative comparison of the performance on the data with or without this unmatching.

---

> ### Author Rebuttal · Authors · 2023-08-10
>
> We thank you for your reviews and address your concerns as follows.
>
> **Q1**: The main task is the “audio-visual” video parsing task, then how would the relationship between the two modalities influence the task?
>
> **A1**: Both modalities contribute to the recognition of events in audio-visual parsing. Even audio signals are still useful in visual event prediction. When we treat audio and visual modality separately, i.e., only using audio signals to predict audio events and only using visual frames to predict visual events, we found the segment-level overall metric drops from 48.9 to 43.1, and the event-level overall metric also decreases by 7.5 points, compared to the model with the interaction and cross attention of audio-visual modalities.
>
> In addition, as can be seen from Table 1 of the manuscript, when we use the Clip visual feature in HAN and MGN, all the metrics including pure audio recognition performance have been improved, which means better visual features could benefit not only visual recognition but also audio recognition.
>
> **Q2**: In default case, would the audio modality perform better than visual modality?
>
> **A2**: Thanks for the question. Audio doesn't always perform better than visual but indeed has better performance in more cases for audio-visual events. Visual frames usually have occlusion, camera movement, low resolution, and more variance, so the event may be hard to see from the visual modality. In contrast, audio signals are relatively easier to capture and clear to hear, thus they are easier to recognize different events.
>
> **Q3**: Compared the results of “A-V” and others with “A”, the improvements are small. If so, enhancing the audio modality may be more effective for this task, while the audio modality in this work is weaken in the process. This is because that there is no proper audio-based large models like CLIP?
>
> **A3**: Thanks for the question. The audio modality was weakened because we only perform visual-track denoising in the manuscript. To better address your concern, we further conduct experiments with a large-scale audio-language pre-trained model called LAION-CLAP [1] for audio denoising, and observe significant improvement in audio-related metrics. LSLD* means we perform both audio and visual label denoise.
> |Method||| Segment-Level||||| Event-Level|||
> | :--: | :--: | :--: | :--: | :--: | :--: | :--: | :--: | :--: | :--: |:--: |
> | | A | V| A-V|Type|Event |A| V |A-V| Type |Event|
> | HAN  |  60.1| 52.9| 48.9 |54.0 |55.4 |51.3| 48.9 |43.0| 47.7 |48.0|
> | LSLD* |  62.7| 67.1| 59.4| 63.1| 62.2| 55.7|64.3| 52.6 |57.6 |55.2|
> |HAN+Clip+Clap| 66.9| 54.3| 50.0| 57.1| 60.2| 59.1| 50.4 |43.9 |51.2 |55.8
> |LSLD*+Clip+Clap| **68.7**| **71.3**| **63.4** |**67.8** |**68.2** |**61.5** |**67.4** |**55.9** |**61.6**| **60.6**
>
> **Q4**: How to generate the prompts? how to obtain A and B? Randomly choose a negative class from 25 categories together with the original class to compose A and B respectively? And the paper says that “the language prompts of the video [could] be xx”. Are there any other formats besides the “other/none/notclass” format in experiment?
>
> **A4**: We apologize that we didn't describe it clearly. Here is an example. Suppose the label annotation of a video is that the video contains Violin (denoted as A) and Cello (denoted as B). Then, we construct the prompts for this video as [Violin and Cello, Violin, Cello, Other]. There is no other format used in the paper. We tried to replace "Other" with "None" and "Notclass" and found "Other" achieves the best performance.
>
> **Limitations**: One main motivation of this paper is to tackle the problem of unmatching between the audio and visual modality. Then what’s the actual situation in the evaluated data? And it would be better to provide statistics of this unmatching in this data to show that the data is proper to evaluate this point, together with a quantitative comparison of the performance of the data with or without this unmatching.
>
> **A5**: Since the training set only has video-level labels, we investigate the unmatching percentage of videos and segments on the validation set. We discover that the unmatching ratio is 73% for videos and 48% for segments, which indicates that unmatching between the audio and visual modality appears extensively in the dataset. Since the training set does not have segment-level labels, we cannot train the model on the matching and unmatching data respectively. However, existing works have studied audio-visual synchronization [2] [3] and Asynchronous [4] [5] [6] phenomenon.
>
> **References**
>
> [1] Yusong Wu, Ke Chen, Tianyu Zhang, Yuchen Hui, Taylor Berg-Kirkpatrick, and Shlomo Dubnov. Large-scale contrastive language-audio pretraining with feature fusion and keyword-to-caption augmentation. In ICASSP, 2023.
>
> [2] Naji Khosravan, Shervin Ardeshir, and Rohit Puri. On attention modules for audio-visual synchronization. In CVPR Workshops, pages 25–28, 2019.
>
> [3] Yasheng Sun, Hang Zhou, Ziwei Liu, and Hideki Koike. Speech2talking-face: Inferring and driving a face with synchronized audio-visual representation. In IJCAI, volume 2, page 4, 2021.
>
> [4] Juergen Luettin, Gerasimos Potamianos, and Chalapathy Neti. Asynchronous stream modeling for large vocabulary audio-visual speech recognition. In 2001 ICASSP, pages 169–172, 2001.
>
> [5] Chuang Gan, Yi Gu, Siyuan Zhou, Jeremy Schwartz, Seth Alter, James Traer, Dan Gutfreund, Joshua B Tenenbaum, Josh H McDermott, and Antonio Torralba. Finding fallen objects via asynchronous audio-visual integration. In CVPR, pages 10523–10533, 2022.
>
> [6] Lee, Sangmin, Sungjune Park, and Yong Man Ro. Audio-Visual Mismatch-Aware Video Retrieval via Association and Adjustment. In ECCV, pages 497-514, 2022.

---

> > ### Comment · Reviewer_Nz2w · 2023-08-18
> >
> > Thank you for the detailed response from the authors. Most of my concerns have been addressed, and I have a few suggestions to the authors. Firstly, regarding the results after incorporating the audio-language pre-trained model and the discussion about enhancing audio modality, they could be included in either the main body of the text or the appendix. Secondly, concerning the design of prompts and the observation that using "other" yields better results than "None" and "Notclass," it would be beneficial to include or briefly mention this in the main text to provide readers with more information.

---

> > > ### Author Response · Authors · 2023-08-18
> > > **Thanks for your suggestions!**
> > >
> > > Thanks for your recognition and suggestion. We will include the above discussions in the revised main text.

---

### Official Review · Reviewer_ULdQ · 2023-07-07

**Soundness:** 3 good
**Presentation:** 4 excellent
**Contribution:** 2 fair
**Rating:** 5
**Confidence:** 5

**Summary:**

This paper proposes LSLD, which leverages CLIP model to denoise unreliable segment-level video labels for audio-visual video parsing.

**Strengths:**

$+$ The proposed method can achieve state-of-the-art results on LLP datasets in several metrics.

$+$ The improvement of video and audio-visual events is significant.

**Weaknesses:**

$-$ The proposed method can only denoise labels for visual domains. It will limit the overall accuracy of audio-visual video parsing.  Although the authors mentioned CLAP model is not applicable to the proposed pipeline in L.59-L.60, the audio waveform can be simply blocked out to get segment-level audio information as well.

$-$ Introducing the language model might not be necessary. Since LLP dataset is collected from AudioSet, pre-trained models from AudioSet can find the corresponding labels for LLP in both audio and visual labels. These models may contribute to stronger label-denoising.

$-$ The effectiveness of LSLD is not clear.
* In L.231, it mentioned that LSLD could also benefit audio accuracy. However, LSLD causes an accuracy drop from 62.4 to 62.3 (Segment-Leve) and from 53.9 to 53.0 (Evet-Level). Since LSLD is based on HAN, which leverages audio-visual feature aggregation, LSLD would be expected to improve audio results as well.
* Does LSLD also benefit other approaches (e.g., MGN)? The proposed approach should also benefit other baselines including CLIP encoder or Resnet encoder.
* Is LSLD complementary to other denoising approaches (i.e., MA and JoMoLD)? These approaches also denoise audio labels. They should be complementary to LSLD.



**Questions:**

$-$ Do the baselines with labels denoising (e.g., MA) also leverage CLIP to do denoise? For example, for MA+CLIP baseline, the refined labels are from CLIP or ResNet?

**Limitations:**

The authors mentioned this in the paper. The proposed method can only work on visual segment-level labels. However, this claim might not be entirely correct. The possible way to do this is mentioned at weakness.

---

> ### Author Rebuttal · Authors · 2023-08-10
>
> We thank you for your reviews and address your concerns as follows. And LSLD means only visual denoise and LSLD* means both audio and visual label denoise in the following answers.
>
> **Q1**: The proposed method can only denoise labels for visual domains.
>
> Now we have extended our denoising method to the audio modality, which leads to further improvement compared to the visual-only denoising results (e.g., from 51.3 to 55.7 on the event-level audio accuracy), even with the VGGish feature.
>
> To be specific, the audio-denoising is based on the audio-text similarity computed by LAION-CLAP [1], which is a large-scale audio-language model pre-trained on Audioset. Same to the denoising process of visual labels (Sec. 3.3), the similarity between prompts and segments is calculated, where the event of the most similar prompt is regarded as the denoised segment-level label.
> |Method||| Segment-Level||||| Event-Level|||
> | :--: | :-: | :-: | :-: | :-: | :-: | :-: | :-: | :-: | :-: |:-: |
> | | A | V| A-V|Type|Event |A| V |A-V| Type |Event|
> | HAN  |  60.1| 52.9| 48.9 |54.0 |55.4 |51.3| 48.9 |43.0| 47.7 |48.0|
> | LSLD* |  62.7| 67.1| 59.4| 63.1| 62.2| 55.7|64.3| 52.6 |57.6 |55.2|
> |HAN+Clip+Clap| 66.9| 54.3| 50.0| 57.1| 60.2| 59.1| 50.4 |43.9 |51.2 |55.8
> |LSLD*+Clip+Clap| **68.7**| **71.3**| **63.4** |**67.8** |**68.2** |**61.5** |**67.4** |**55.9** |**61.6**| **60.6**|
>
>  'Clip + Clap' means we use the visual feature extracted from CLIP and the audio feature extracted from LAION-CLAP.
>
> **Q2**: Introducing the language model might not be necessary.
>
> The AudioSet pre-trained model learns from audio-visual correlations, making it more suitable for handling videos whose audio and visual events are temporally. However, in AVVP, audio and visual are assumed to be misaligned, and the model needs to predict different labels for the two modalities. Thus the AudioSet pre-trained model cannot be directly used for label denoising. Also, it cannot be used to generate segment-level labels.
>
> In contrast, a language model is more general but effective way to denoise the audio and visual tracks individually. Introducing CLIP or CLAP is crucial, since language models are flexible and can describe all events occurring in a video, enabling the generation of segment-level labels. Furthermore, using CLIP makes our method capable of extending to other datasets and tasks, not confined solely to a subset of AudioSet. Moreover, our audio backbone (VGGish) is pre-trained on AudioSet, but using CLIP still leads to better performance upon it.
>
> **Q3**:  LSLD would be expected to improve audio results as well.
>
> Since the original LSLD only performs visual denoising, the performance of audio accuracy is not that highly assured. Although in Table 1 of the manuscript, the audio accuracy is slightly decreased, LSLD indeed improves the audio accuracy upon DHHN[3] and MM-Pyramid[2] as shown in Sec. A of the appendix. Please see the table below.
> |Method||| Segment-Level||||| Event-Level|||
> | :-: | :-: | :-: | :-: | :-: | :-: | :-: | :-: | :-: | :-: |:-: |
> | | A | V| A-V|Type|Event |A| V |A-V| Type |Event|
> |  MM-Pyramid  |   60.9 |54.4 |50.0 |55.1 |57.6 |52.7 |51.8 |44.4 |49.9 |50.5|
> | MM-Pyramid + Clip | 62.8 |56.3 |51.6 |56.9 |60.0 |53.1 |52.2 |45.5 |50.3 |50.7|
> |MM-Pyramid + LSLD + Clip| **63.3** |**66.8** |**60.4** |**63.5**| **62.4**|**54.5**|**62.5**|**52.6**|**56.5**| **53.2**|
> |  DHHN  | 61.3| 58.3 |52.9 |57.5 |58.1 |54.0 |55.1 |47.3 |51.5 |51.5|
> | DHHN + Clip |  62.6 |59.1 |53.4 |58.4 |59.1 |54.3| 55.2| 46.8| 52.1| 52.5|
> |DHHN + LSLD + Clip|**64.1**|**69.0**|**60.7**|**64.6**|**64.0**|**56.2**|**66.2**|**53.7**|**58.7**|**56.4**|
>
> In addition, we also performed LSLD* by extending our method into audio denoising, which leads to significant improvement over audio metrics. Please refer to the table of Q1.
>
> **Q4**: Does LSLD also benefit other approaches? The proposed approach should also benefit the CLIP encoder or Resnet encoder.
>
> In the original supplementary files, we reported the results of applying LSLD on MM-Pyramid and DHHN. The results are shown in the table of Q3. We can see LSLD clearly improves both architectures.
>
> Following your suggestion, we also tested on the ResNet encoder with LSLD*, please see the results of 'LSLD*' from the table of Q1.
>
> **Q5**: Is LSLD complementary to other denoising approaches?
>
> Yes, we apply LSLD* to MA and JoMoLD and see both of them obtain significant improvement. All the experiments are conducted with the VGGish feature and Resnet feature for a fair comparison. Please see the table below.
> |Method||| Segment-Level||||| Event-Level|||
> | :-------: | :-: | :-: | :-: | :-: | :-: | :-: | :-: | :-: | :-: |:-: |
> | | A | V| A-V|Type|Event |A| V |A-V| Type |Event|
> |  MA  |   60.3| 60.0| 55.1| 58.9| 57.9 |53.6| 56.4| 49.0| 53.0| 50.6|
> |MA + LSLD* | **61.2**| **66.6**| **59.0**| **62.3**| **61.1**| **54.3**| **64.0**| **52.5**| **56.9**| **54.2**|
> |  JoMoLD  |  61.3 |63.8| 57.2| 60.8| 59.9| 53.9| 59.9| 49.6| 54.5| 52.5|
> |JoMoLD + LSLD* | **62.3**| **66.1**| **58.7**| **62.4**| **61.8**| **55.8**| **63.6**| **52.2**| **57.2**| **55.0**|
>
> **Q6**: Do the baselines with labels denoising also leverage CLIP to do denoise? For MA+CLIP baseline, the refined labels are from CLIP or ResNet?
>
> No, MA and JoMoLD don't leverage CLIP to denoise labels. For MA+CLIP, the refined labels are from CLIP visual features, but CLIP similarity (e.g., computing similarity between the image and our constructed prompt) is not used in the denoising process.
>
> [1] Yusong Wu et al. Large-scale contrastive language-audio pretraining with feature fusion and keyword-to-caption augmentation.
>
> [2] Jiashuo Yu et al. Mm-pyramid: Multimodal pyramid attentional network for audio-visual event localization and video parsing.
>
> [3] Xun Jiang et al. Dhhn: Dual hierarchical hybrid network for weakly-supervised audio-visual video parsing.

---

> > ### Comment · Reviewer_ULdQ · 2023-08-14
> >
> > Thanks for providing the response. It addressed my main concern. Thus, I increase my rating. Also, I encourage the authors to include baselines (mentioned in 2DCt response [1-8] ) in Table 1.

---

> > > ### Author Response · Authors · 2023-08-15
> > > **Thank you for recognizing our responses!**
> > >
> > > Thanks for recognizing our responses. We will definitely include these baselines in our revision. Thank you for helping to improve our work again!

---

### Official Review · Reviewer_Q15u · 2023-07-09

**Soundness:** 3 good
**Presentation:** 3 good
**Contribution:** 3 good
**Rating:** 5
**Confidence:** 4

**Summary:**

The authors have observed that weakly-supervised labels in audio-visual tasks are noisy at the segment-level label. To solve the problem, the authors introduce language as an additional source of information to assign soft labels to each of the 1-second video segments. More specifically, a similarity score is computed between each segment and possible prompts, and segment-level labels are softly assigned by the proposed dynamic re-weighting method. The experimental results show the performance boost with the re-weighing.

**Strengths:**

1. The paper is mostly well-written, with details and figures.

2. The idea of dynamic re-weighting for segment-level label noise is interesting.

3. The authors show various experimental results, including the ablation study of different prompts.

**Weaknesses:**

1. Some manuscripts need to be clarified.

* Section 3.4, especially Equation (4), needs to be clarified. According to Equation (4), $\tilde{\mathbf{y}}_t^v[c]$ is already assigned with 1 or 0. Where does the value come from?

* Moreover, there is a discrepancy between the descriptions in Lines 171-179 and the equation. For example, how is *relatively high similarity* applied to the equation?

* Figure 1 (c): Does```w/o Cello``` stand for a a prompt "w/o Cello"?

* In Table 1's caption, it is written, "The last 3 lines are all label denoising methods ..."
Do the last 3 lines indicate ```MA+Clip```, ```JoMoLD+Clip```, and ```LSLD (ours)```?

2. Although Section 4 shows multiple experimental results, all of the results are on a single dataset (LLP dataset). Ideally, the proposed approach should be demonstrated on various datasets, especially those with no segment-level labels or longer video clips.

**Questions:**

1. Does ```other``` include the case where the segment is similar to some other labels than class A or class B?

**Limitations:**

The authors discussed the limitations of the work (e.g. segment-level labels for audio) in Section 5.

---

> ### Author Rebuttal · Authors · 2023-08-09
>
> We thank you for your reviews and address your concerns as follows.
>
> **Q1**: Section 3.4, especially Equation (4), needs to be clarified. According to Equation (4), $\mathbf{\widetilde{y}}_t^v[c]=1$is already assigned with 1 or 0. Where does the value come from?
>
> **A1**:  Sorry for the confusion. The value is derived from Sec 3.3, where each visual segment is assigned with segment-level label $\mathbf{\widetilde{y}}_t^v$ by segment-level denoising. After denoising, for each probable class $c$, $\mathbf{\widetilde{y}}_t^v[c]$ could be either 0 (not exist) or 1 (exist). An example of the label denoising process is illustrated below:
>
> Suppose we have a video containing two events, Violin and Cello, its video-level label is assigned as $\mathbf{y}^v$['Violin']=1, $\mathbf{y}^v$['Cello']=1. The labels of each segment are initially the same as the video-level label. To perform label denoising, the prompts are created as [Violin and Cello, Violin, Cello, Other], where ``Other'' means neither Violin nor Cello appears. Then we calculate the similarity between segments and prompts, and take the prompt with the highest similarity as the event that appears in this segment. Suppose for $t$-th segment, the prompt with the highest similarity is Violin,
> we then adjust $\mathbf{\widetilde{y}}_t^v$['Cello'] to 0, while retaining $\mathbf{\widetilde{y}}_t^v$['Violin'] at 1. In this way, we obtain the segment-level denoised label $\mathbf{\widetilde{y}}_t^v[c]$ for each video segment.
>
> **Q2**: There is a discrepancy between the descriptions in Lines 171-179 and the equation. For example, how is relatively high similarity applied to the equation?
>
> **A2**: We apologize that we didn't describe it clearly. Lines 171-179 describe the procedure of finding unreliable segments, after that we apply dynamic reweighting to these unreliable segments by Eq.(4).
>
> For positive samples ($\\mathbf{\\widetilde{y}}_t^v[c] = 1$), the unreliable one is that its prompt similarity score $\\mathbf{s}_c$ is lower than the maximum similarity scores of negative data ($Max_w/o$). For negative samples, the unreliable one is that its prompt similarity score $\\mathbf{s}_c$ is higher than the minimum similarity scores of positive data ( $Min_w$ ). Indeed the unreliable data are the intersection of the yellow region (negative data) and blue region (positive data) as shown in Fig. 2 of the original manuscript. We then change the labels of unreliable samples by a soft value (proportional to its prompt similarity), rather than the hard values (0 or 1). For those reliable ones, we keep their original label unchanged.
>
> Thus the detailed procedure for applying dynamic re-weighting on unreliable segments is illustrated below,
>
> $$\mathbf{\hat{y}}_t^v[c]=
> \begin{cases}
> Min(1,\alpha \times \mathbf{s_c} ) & ,\mathbf{\widetilde{y}}_t^v[c] = 1 \\ and \\  \mathbf{s_c} \leq Max_w/o \\\\
> Min(1,\beta \times \mathbf{s_c})&, \mathbf{\widetilde{y}}_t^v[c] = 0 \\ and \\ \mathbf{s_c} \geq Min_w\\\\
> \widetilde{y}_t^v[c]& , otherwise
> \end{cases}$$
>
> where $ \mathbf{s_c}$ is the similarity between the segment and the event, $Min_w$ is the lowest similarity of the segments with the event, $Max_w/o$ is the highest similarity of segments w/o the event, $\alpha$ and $\beta$ are two scalars.
>
> **Q3**: Does w/o Cello stand for a prompt "w/o Cello"?
>
> **A3**: As illustrated in Q2, w/o Cello stands for segments that are labeled without the Cello event. We will make it clearer in our manuscript.
>
> **Q4**: In Table 1's caption, it is written, "The last 3 lines are all label denoising methods ..." Do the last 3 lines indicate MA+Clip, JoMoLD+Clip, and LSLD (ours)?
>
> **A4**: Yes, MA, JoMoLD, and LSLD(Ours) are all label denoising methods. And MA+Clip, JoMoLD+Clip means that we reproduce MA and JoMoLD with extracted features from Clip.
>
> **Q5**: Although Section 4 shows multiple experimental results, all of the results are on a single dataset (LLP dataset). Ideally, the proposed approach should be demonstrated on various datasets, especially those with no segment-level labels or longer video clips.
>
> **A5**: Sorry that we have not carry out experiments on additional datasets. Since LLP is the only dataset that has both segment-level and modality-level labels for test, all state-of-the-art AVVP methods were evaluated exclusively on the LLP dataset. In the future, we will try to conduct experiments on more general datasets.
>
> **Q6**: Does 'other' include the case where the segment is similar to some other labels than class A or class B?
>
> **A6**: Yes, `Other' means that neither class A nor class B appears.

---

> > ### Author Response · Authors · 2023-08-19
> > **Welcome to discuss!**
> >
> > Dear reviewer,
> >
> > We are following up on our paper rebuttal. In summary, we elaborated on deriving the value of $\mathbf{\widetilde{y}}_t^v[c]$ and modifying Eq. 4 to incorporate *relatively high similarity*. We explained the meanings of 'w/o Cello' and 'other', and emphasized the reason for experimenting solely on the LLP dataset. Your response and feedback are highly valued. Thank you for your time and consideration.

---

> > > ### Comment · Reviewer_Q15u · 2023-08-20
> > >
> > > Thanks for the rebuttal. The authors addressed my concerns, and I have raised my rating from 4: Borderline reject to 5: Borderline accept.

---

> > > > ### Author Response · Authors · 2023-08-20
> > > > **Thanks for your comment!**
> > > >
> > > > Dear Reviewer Q15u, thanks for recognizing our work. We are happy that our response has addressed your concerns. We will include these clarifications in our final version.

---

### Decision · Program_Chairs · 2023-09-21

**Decision:**

Accept (poster)

**Comment:**

Initially, all four expert reviewers appreciated the paper’s contribution. They appreciated the simplicity of the approach and the demonstrated improvement over the prior art. The rebuttal addressed the reviewers’ concerns. All four reviewers recommended that the paper is above the bar for acceptance to NeurIPS. This AC agrees with their recommendation. Please take into account all reviewer feedback in the camera-ready version.